# Transcriptomic Signature and Pro-Osteoclastic Secreted Factors of Abnormal Bone-Marrow Stromal Cells in Fibrous Dysplasia

**DOI:** 10.3390/cells13090774

**Published:** 2024-04-30

**Authors:** Zachary Michel, Layne N. Raborn, Tiahna Spencer, Kristen S. Pan, Daniel Martin, Kelly L. Roszko, Yan Wang, Pamela G. Robey, Michael T. Collins, Alison M. Boyce, Luis Fernandez de Castro

**Affiliations:** 1Metabolic Bone Disorders Unit, National Institute of Dental and Craniofacial Research, National Institutes of Health, Bethesda, MD 20892, USA; zachary.michel@nih.gov (Z.M.); kpan4@jhmi.edu (K.S.P.); alison.boyce@nih.gov (A.M.B.); 2Skeletal Diseases and Mineral Homeostasis Section, National Institute of Dental and Craniofacial Research, National Institutes of Health, 30 Convent Drive, Building 30, Room 207, Bethesda, MD 20892, USA; laynenr123@gmail.com (L.N.R.); ashley23spencer@gmail.com (T.S.); kelly.roszko@nih.gov (K.L.R.); mcollins@nih.gov (M.T.C.); 3Genomics and Computational Biology Core, National Institute of Dental and Craniofacial Research, National Institutes of Health, Bethesda, MD 20892, USA; daniel.martin@nih.gov; 4Mass Spectrometry Facility, National Institute of Dental and Craniofacial Research, National Institutes of Health, Bethesda, MD 20892, USA; yan.wang2@nih.gov; 5Skeletal Biology Section, National Institute of Dental and Craniofacial Research, National Institutes of Health, Bethesda, MD 20892, USA; probey@dir.nidcr.nih.gov

**Keywords:** fibrous dysplasia, bone-marrow stromal cell, osteoprogenitor, skeletal stem cell, cytokines, skeleton, osteoclast

## Abstract

Fibrous dysplasia (FD) is a mosaic skeletal disorder caused by somatic activating variants of *GNAS* encoding for Gα_s_ and leading to excessive cyclic adenosine monophosphate signaling in bone-marrow stromal cells (BMSCs). The effect of Gα_s_ activation in the BMSC transcriptome and how it influences FD lesion microenvironment are unclear. We analyzed changes induced by Gα_s_ activation in the BMSC transcriptome and secretome. RNAseq analysis of differential gene expression of cultured BMSCs from patients with FD and healthy volunteers, and from an inducible mouse model of FD, was performed, and the transcriptomic profiles of both models were combined to build a robust FD BMSC genetic signature. Pathways related to Gα_s_ activation, cytokine signaling, and extracellular matrix deposition were identified. To assess the modulation of several key secreted factors in FD pathogenesis, cytokines and other factors were measured in culture media. Cytokines were also screened in a collection of plasma samples from patients with FD, and positive correlations of several cytokines to their disease burden score, as well as to one another and bone turnover markers, were found. These data support the pro-inflammatory, pro-osteoclastic behavior of FD BMSCs and point to several cytokines and other secreted factors as possible therapeutic targets and/or circulating biomarkers for FD.

## 1. Introduction

Fibrous dysplasia (FD) is a mosaic skeletal disorder caused by somatic activating variants in *GNAS*, encoding the α subunit of the stimulatory G protein. The location and extent of bone lesions are variable and can be associated with hyperpigmented skin macules and/or various endocrinopathies, termed McCune–Albright syndrome (MAS, OMIM #174800). Although MAS is a rare disorder (prevalence of 1:100,000 to 1:1,000,000), FD accounts for as much as 7% of all benign bone tumors [1]. GNAS p.R201C and p.R201H activating variants are typically the genetic causes. These mutations lock Gα_s_ in an active conformation, leading to the excessive production of intracellular cyclic adenosine monophosphate (cAMP) in bone-marrow stromal cells (BMSCs). In bone, this results in FD lesions that become apparent during early childhood and can lead to deformity, fractures, and pain [2]. Due to the mosaic nature of FD, disease burden can range from a single lesion (monostotic) to multiple areas of the skeleton (polyostotic), leading to severe disability in the patients [3].

FD arises from the altered differentiation of bone-marrow skeletal stem cells (SSCs), the progenitor subpopulation of BMSCs [4]. Affected SSCs are unable to differentiate into either hematopoiesis-supporting marrow stroma or adipocytes, instead generating highly proliferative fibroblastic cells and abnormal osteochondrogenic cells. This leads to the regression of normal hematopoietic/fatty marrow tissue into the perimeter of the lesions and its replacement by fibro-osseous tissue with occasional areas of fibrocartilage. Radiographically, lesions typically appear as lytic/sclerotic “ground glass” areas within the bones, although appearances can vary significantly [5]. On histology, they appear as vascularized fibro-osseous tissue of undifferentiated fibroblasts and curvilinear trabeculae of abnormal, poorly mineralized woven bone [5].

FD lesions are often rich in osteoclasts and osteolytic activity, and an elevated bone turnover is a characteristic of the disease [6]. Indeed, osteoclasts are recruited by an abnormally high production of RANKL by FD BMSCs such that circulating RANKL in patients with FD is 16-fold higher compared to healthy controls [7]. These findings led to the consideration of RANKL inhibition as a therapeutic approach for FD [8,9]. In a clinical trial conducted by our group, the anti-RANKL drug denosumab successfully halted bone resorption in FD and prevented the proliferation of altered BMSCs, leading to the decreased cellularity and normalized differentiation of affected BMSCs. This resulted in improved mineralization and organization of bone in FD lesions and a decrease in co-morbidities associated with the skeletal lesions. Based on these findings, our current understanding of FD pathophysiology is that of a positive feedback loop between altered BMSCs and osteoclasts. In this scenario, FD BMSCs release factors inducing osteoclastogenesis, and osteoclasts respond by releasing RANK-containing extracellular vesicles to increase BMSC proliferation and further altering their differentiation [10,11]. Despite this advancement, little is known about the transcriptional effects of Gα_s_ activation/cAMP excess in BMSCs and how this translates to changes within the FD microenvironment.

Therefore, a comprehensive characterization of the genetic expression and production of secreted factors in addition to RANKL and OPG is crucial to understand the microenvironment of FD lesions. Moreover, as the discovery of RANKL’s role in FD led to the trial of denosumab therapy, such an exploration would provide novel therapeutic approaches and/or circulating biomarkers for FD. Previous efforts have been taken to characterize other secreted factors in FD, including the demonstration of lesional FGF23 excess [12] and investigations of IL-6 that led to an unsuccessful attempt to treat FD patients with the IL6R-inhibiting drug tocilizumab [13,14,15]. However, the scope of these exploratory studies was limited due to the lack of high-throughput screening techniques. 

On the other hand, we and others carried out efforts to characterize the transcriptomic profile of FD. Our approach involved isolating bulk mRNA from FD tissue; while informative, this did not allow us to determine how *GNAS* variants independently affect gene expression in BMSCs [11,16,17,18]. In addition, previous studies characterized the transcriptomic changes caused by Gα_s_ activation using cultured human BMSCs transduced with Gα_s_^R201C^, though the substantial cellular manipulation of this technique may limit its reliability to capture the transcriptomic effects of the Gα_s_ gain of function in lesional BMSCs [19,20].

We developed a methodology that utilized a combination of bulk RNAseq performed with cultured BMSCs from an inducible mouse model of FD and cultured BMSCs of FD patients compared with healthy volunteers. This resulted in the development of a robust transcriptomic signature specific to altered FD BMSCs, including only genes significantly and similarly modulated in both systems. We then measured the concentration of several pro-osteoclastogenic cytokines and other factors of interest secreted into culture media. Lastly, we measured the circulating concentrations of some of these cytokines in plasma samples from patients with FD and correlated them with disease burden, resulting in the proposal of novel potential circulating biomarkers for FD.

## 2. Materials and Methods

### 2.1. Human and Mouse Specimens

Primary cultures of bone lesions from six patients with FD, as well as plasma samples from fifty-seven patients with FD, were selected from a longstanding natural history protocol (NCT00001727) (Appendix A, Appendix A). The protocol was approved by the NIH Investigational Review Board, and all subjects/guardians gave informed consent/assent. Subjects were limited to those who had never received bisphosphonate or denosumab therapy. All subjects had skeletal disease burden scores (SBSs) measured using a previously validated Tc-99m bone scintigraphy method [21]. Plasma was collected from patients at NIH and cryopreserved; bone specimens were collected as waste during corrective surgeries and used to culture primary BMSCs, which were subsequently cryopreserved. BMSCs from six healthy donors were obtained according to NIH ethical guidelines (NIH OHSRP exemption #373) (Appendix A).

Femora and tibiae from 4 inducible FD mice carrying a doxycycline-inducible Gα_s_^R201C^ variant were dissected and used to obtain primary murine BMSC cultures (Appendix A). To limit the variability introduced by mixing sexes and ages in the analyses, all mice used were 12-week-old males.

### 2.2. Human BMSC Culture

Cryopreserved BMSCs isolated from 16 patients with FD and 6 healthy volunteers (HV) were cultured as previously described [22] and cryopreserved. Cultures were thawed and seeded in T-25 flasks for continued growth in full medium (α-MEM supplemented with 20% fetal bovine serum and penicillin/streptomycin) until at least 5.5 million cells were present. Cells were detached using trypsin (Sigma-Aldrich, St. Louis, MO, USA, T4049) and the number of total cells was confirmed using a Cellometer Auto M10 (Nexcelom Bioscience, Lawrence, MA, USA). Two T-25 flasks were seeded with 2 million BMSCs in full medium for each cell line; after one day, medium was changed to depleted medium (α-MEM supplemented with penicillin/streptomycin). Cells were then incubated for 48 h, then culture medium was collected from each well. TRIzol Reagent (Invitrogen, Waltham, MA, USA, 15596026) was added to flasks and frozen at −80 °C. Complete Mini EDTA Free Protease Inhibitor Cocktail (Roche, Basel, CH, 4693159001) was added to medium, which was centrifuged to discard aggregates and concentrated 10 times using Amicon Ultra-4 Ultracel-3 3 kDa centrifugal filter units (Millipore, Burlington, MA, USA, UFC800308). Aliquots of concentrated medium and flowthrough medium were frozen at −80 °C.

### 2.3. Murine BMSC Culture

To avoid excessive presence of hematopoietic cells in the primary cultures, mouse bones were depleted of bone marrow, minced into small chips, and seeded onto T-25 flasks using full medium (same formulation as in human cultures). Medium was changed after 6 h and then every 4 days. Bone chips were subsequently removed and placed in new T-25 flasks on days 12 and 21. On day 28, flasks showing high ratio of stromal cells to monocytes as per visual examination through an inverted microscope were treated with trypsin (Sigma Aldrich, T4049) and harvested cells were transferred to T-75 flasks. When an estimated 12 million cells per subculture were achieved, cells were trypsinized and passed through a nylon mesh for single-cell suspension. The total number of live cells was confirmed using a Cellometer Auto M10 (Nexcelom Bioscience) and Trypan Blue (Sigma-Aldrich, T8154). Cultures were negatively immunoselected for hematopoietic cells using CD45 and CD11b microbeads (Miltenyi Biotec, Gaithersburg, MD, USA, 130-052-301 and 130-097-142) and the autoMACS Pro Separator following the manufacturer instructions. Six-well plates were seeded with 0.8 million cells per well in full medium. After 48 h, the medium was changed for depleted medium (same formulation as in human cultures). Half of the cultures from each mouse were treated with doxycycline hyclate (5 mg/mL, Sigma-Aldrich D5207) for 48 h and media was collected. Complete Mini EDTA Free Protease Inhibitor Cocktail (Roche, 4693159001) was added to media. Cells were then incubated for 48 h, then culture medium was collected from each well. TRIzol Reagent (Invitrogen, 15596026) was added to flasks and frozen at −80 °C. Media was centrifuged to discard aggregates and concentrated 10 times using Amicon Ultra-4 Ultracel-3 3 kDa centrifugal filter units (Millipore, UFC800308). Aliquots of concentrated media and flowthrough media were frozen at −80 °C.

### 2.4. RNA Extraction, Sequencing, and Analysis

Bulk RNA was extracted from cultured cells using the TRIzol Reagent protocol without glycogen. Library preparation and sequencing was performed by the Genomics and Computational Biology Core (NIDCR, NIH) team using an Illumina HiSeq 2500 (Illumina, San Diego, CA, USA) configured for 37 paired-end reads (human) or 150 paired-end reads (mouse). Read quality was assessed using FastQC software v0.12.0 [23] and reads were subsequently mapped using STAR aligner v2.5.2a [24]. Mapping quality was assessed using Picard tools. Read counts were determined using the quantMode utility of STAR aligner and genes with less than 5 counts in at least 1 sample were filtered out. Normalized counts were then calculated using TMM normalization. Cultures showing GNAS p.R201C/H mRNA variant allele expression frequency of 7–55% were selected for further analyses (6 out of 16 samples cultured). Briefly, bam alignments were visualized using the Integrative Genome Viewer (IGV) application and nucleotide counts at the GNAS p.R201 position and corresponding amino acid changes were determined for each sample [25]. DEseq2 [26] was used to determine differential gene expression. Unsupervised clustering was performed using principal component analysis and unsupervised clustering using heatmap.2 in R [27]. Other heatmaps were manually generated by grouping related genes that were differentially regulated in the same direction.

Cross-species human–mouse annotation was performed using the Biomart service using the high-confidence ortholog dataset. Annotated genes were then used to determine the number of genes differentially regulated in the same manner between both species (defined as the FD Signature). Pathway analysis was performed for humans, mice, and the FD Signature using Enrichr (https://maayanlab.cloud/Enrichr/ [accessed 18 April 2023]) to generate bar charts from GO Molecular Function 2021, BioPlanet 2019, KEGG 2021 Human, and MSigDB Hallmark 2020 using |log_2_-fold change| > 1.5 and adjusted *p*-value < 0.01.

Predicted protein–protein interactions were mapped based on genes differentially regulated in the same direction in both species (adjusted *p* < 0.01) using the STRING database (v11.5; string-db.org). The top 6 groups resulting from MCL clustering were annotated based on common protein functions.

Sequencing datasets are available at the NCBI GEO repository under series accession number GSE261360.

### 2.5. cAMP Determination

cAMP-Gs HiRange standard (Cisbio, Codolet, FR, 62AM6CDA) was dissolved in 11.2 mM 3-isobutyl-1methylxanthine (IBMX; Sigma-Aldrich, I5879) and Dulbecco’s Modified Eagle Medium (DMEM; Gibco, Leicestershire, UK, 10564011) for external calibration. A 200 µL aliquot of post-filtration flowthrough media was combined with 500 µM IBMX to protect the setup from cAMP degradation. The aliquot was combined with 200 µL acetonitrile and centrifuged (13,000× *g*, 30 min); 50 µL of supernatant was combined with 50 µL HPLC-grade water.

Liquid chromatography with tandem mass spectrometry was then used to quantify cAMP using an orbitrap Fusion Lumos mass spectrometer interfaced to an Ultimate3000 HPLC system (Thermo Scientific, West Palm Beach, FL, USA) and an Atlantis C18 column (2.1 × 150 mm, 3 µm, Waters Corp, Milford, MA, USA). Samples were injected and desalted for 2 min in 2% solvent B (80% acetonitrile, 20% water, and 0.1% formic acid) and in solvent A (0.1% formic acid in water), followed by a linear gradient to 98% B in 3 min; the composition was held at 98% B for 2.5 min before ramping down to 2% B in 0.1 min. Column was equilibrated for 4.5 min at 2% B before the next injection. Flow rate was 250 µL/min. Solvent A and solvent B were Optima-grade from Fisher Scientific. Parallel Reaction Monitoring of protonated cAMP at *m*/*z* 330.0598 ([M + H]^+^) was recorded in the orbitrap at R = 50,000 for detection and quantification using positive-ion electrospray ionization with the following source parameters: spray voltage—3300 V, sheath gas—40 units, aux gas—10 units, and sweep gas—2 units. The ion transfer tube was at 325 °C and vaporizer was at 350 °C. Isolation window was 1.2 *m*/*z*; collision energy was 35%. Standard curves were acquired before and after the sample set. A QC sample (5 nM standard) was analyzed mid-sequence. Quantification was carried out using QuanBrowser module of Xcalibur software v4.3 (Thermo Scientific) using peak area of *m*/*z* 136.0609. Quantification result was manually inspected before being exported to Excel (Microsoft, Redmond, WA, USA).

### 2.6. Serum and Media Determinations

For mouse samples, RANKL was detected using Mouse TRANCE/RANKL/TNFSF11 Quantikine ELISA Kit (R&D systems, MTR00); Ephrin B2, Semaphorin 3A, and FAPα using ELISAs from LSBio (LS-F6974, LS-F33608, and LS-F49804, respectively); and soluble Fas ligand and M-CSF using R&D Quantikine assays (MFL00 and MMC00B, respectively). In addition, a customized magnetic Luminex assay was designed to assess Dkk-1, IL-7, MMP-2, β-NGF, and VEGF in mice (R&D Sytems, LXSAMSM-05 with BR77, BR14, BR37, BR43, and BR21). Mouse cytokines were detected with the Bio-Plex Pro Mouse Cytokine 23-plex Assay (Bio-Rad, #M600009RDPD), including IL-1α, IL-1β, IL-2, IL-3, IL-4, IL-5, IL-6, IL-9, IL-10, IL-12 (p40), IL-12 (p70), IL-13, IL-17A, Eotaxin, G-CSF, GM-CSF, IFN-γ, KC, MCP-1 (MCAF), MIP-1α, MIP-1β, RANTES, and TNF-α.

In human samples, RANKL was detected using Biomedica Immunoassay for free sRANKL ELISA kit (Eagle Biosciences, BI-20462), and cytokines were detected using the Bio-Plex Pro Human Cytokine 17-plex Assay (Bio-Rad, #M5000031YV), including IL-1β, IL-2, IL-4, IL-5, IL-6, IL-7, IL-8, IL-10, IL-12 (p70), IL-13, IL-17A, G-CSF, GM-CSF, IFN-γ, MCP-1 (MCAF), MIP-1β, and TNF-α. 

Analytes that were undetectable in human or mouse experimental and control groups were excluded from analyses.

### 2.7. Statistical Analysis

Comparisons between released cytokines in cell cultures were performed on GraphPad Prism v9.2.0 (GraphPad Software, Boston, MA, USA) and determined using multiple Mann–Whitney tests with a two-stage Benjamini, Krieger, and Yekutieli step-up and 5% False Discovery Rate (FDR). The correlations between circulating cytokines and skeletal burden score in patients with FD/MAS were determined using multiple correlations with Pearson’s r in GraphPad. Using SAS v9.4 (SAS Institute, Cary, NC, USA), *p*-values in SBS–cytokine and cytokine-cytokine correlations were adjusted for multiple testing, with an FDR-adjusted *p*-value below 0.05 considered statistically significant. Unpaired *t*-tests were used to determine significance of relationships between cAMP and variant burden comparisons with a threshold of *p* < 0.05. Statistical analysis of differential gene expression was included in the DESeq2 package.

## 3. Results

### 3.1. Murine and Human Gα_s_^R201C/H^-Expressing BMSC Cultures Display Pro-Inflammatory Transcriptomic Profiles

Bone chips from both femora and tibiae from each mouse were minced and plated. Two to four weeks later, cultures were depleted of CD45+ CD11b+ hematopoietic cells and seeded in 6-well plates. Two days later, half of the cultures were induced for Gα_s_^R201C^ expression for 48 h with doxycycline (5 mg/mL), and then mRNA was extracted (Appendix A). Gα_s_^R201C^ transgene expression and downstream pathway activation were confirmed with transcriptomic analysis (Figure 1A) and through the measurement of cAMP in the culture media, respectively (Figure 1B). RNAseq showed good segregation by Gα_s_^R201C^ expression status based on principal component (PC) analysis, which also grouped together on unsupervised clustering (Figure 1C,D) and revealed the significant modulation of 5852 genes (Appendix A). Pathway analysis of the most differentially expressed genes in this dataset (with at least |log2 fold change| > 1.5 and adjusted *p* < 0.01) was performed using the Gene Ontology (GO) Molecular Function, BioPlanet, Kyoto Encyclopedia of Genes and Genomes (KEGG), and Molecular Signatures Database (MSigDB) hallmark gene set repositories with the Enrichr tool. Some of these gene sets showed signatures directly or indirectly associated to Gα_s_ activation, and all of them showed terms related to cytokine signaling and inflammation: “cytokine activity” in the GO Molecular Function; “TGF-beta regulation of extracellular matrix” and “Interleukin-1 regulation of extracellular matrix” in BioPlanet; “Cytokine-cytokine receptor interaction” in KEGG; and “TNF-alpha Signaling via NF-kB” and “Inflammatory Response” in MSigDB (Figure 1E).

Of all sixteen human FD cultures tested, six had a p.R201H or p.R201C variant allele frequency between 7–55% of all *GNAS* reads and were selected for further analyses and compared to BMSCs obtained from healthy volunteers (Figure 2A). Gα_s_ activation was confirmed through the measurement of cAMP in the culture media (Figure 2B). PC analysis showed the excellent segregation of FD and healthy volunteer (HV) BMSCs, and both categories partially grouped together in unsupervised clustering analysis (Figure 2C,D) and revealed the significant modulation of 3637 genes (Appendix A). We performed a similar Enrichr pathway analysis as in mouse cultures, with the same fold change and *p*-value restrictions, which also showed pathways consistent with Gα_s_ activation, and all showed gene sets related to cytokines and inflammation: “cytokine activity” and “chemoattractant activity” in the GO Molecular Function; “TGF-beta regulation of extracellular matrix”, “Cytokine-cytokine receptor interaction”, “Interleukin-4 regulation of apoptosis”, and “TGF-beta regulation of skeletal system development” in BioPlanet; “Cytokine-cytokine receptor interaction” and “TGF-beta signaling pathway” in KEGG; and “TNF-alpha Signaling via NF-kB” and “Inflammatory Response” in MSigDB (Figure 2E).

### 3.2. Combining the Transcriptome of Human and Murine FD BMSCs Reveals a Robust Genetic Signature

To develop a robust signature of dysregulated genes in FD BMSCs across disease models, we combined the datasets of differentially expressed human and mouse genes. Of the 3367 differentially expressed genes in human samples, 2964 had a mouse ortholog, and of the 5852 differentially expressed mouse genes, 5355 had a human ortholog. Of these, 1239 genes appeared in both datasets and 764 were regulated in the same direction (Figure 3A, Appendix A). A selection of these genes involved in pathways of interest for the study of FD, curated through literature review, are shown in Figure 3B, including genes involved in Gα_s_/cAMP signaling, SSC differentiation, fibrosis, inflammation (including pro-osteoclastogenic cytokines), and vascularization. In addition, the 136 genes with strongest regulation (defined by |log_2_ fold change| > 1.5 and adjusted *p* < 0.01) were analyzed with Enrichr pathway analysis, showing over-representation of genes related to cAMP signaling in the GO Molecular Function; “G alpha s pathway” in BioPlanet; “Cytokine-cytokine receptor interaction” in KEGG; and “TNF-alpha Signaling via NF-kB” and “Inflammatory Response” in MSigDB (Figure 3C). Lastly, we selected those genes with *p* < 0.01 (435 genes) to assess the pathway interaction of their encoded proteins using STRING, which detected six main interacting pathways, involved in processes like cAMP signaling, extracellular matrix organization, metallopeptidase activity, cellular metabolism, mesenchymal cell proliferation, and ATPase activity (Appendix A).

### 3.3. Mouse Gα_s_^R201C^-Expressing BMSC Cultures Release Pro-Inflammatory Cytokines and Other Factors Related to FD Pathophysiology

We measured the levels of 18 cytokines in the primary culture media of BMSCs from FD patients and healthy volunteers (HV). Of these, five cytokines were undetectable and, due to the high variability of these samples, we failed to detect significantly different levels of any of the twelve remaining cytokines between HV and FD BMSC culture media (Appendix A). However, we measured twenty-four cytokines in media from murine BMSC cultures, of which thirteen factors were significantly higher in Gα_s_^R201C^-expressing BMSCs (Figure 4), nine did not change (Appendix A), and two fell below the detection range of the assays (IL-4 and IL-9). In addition to these cytokines, we measured other factors involved in the FD microenvironment such as the proteases MMP2 and FAPα, the WNT modulator Dkk1, the growth factors VEGF and β-NGF, and modulators of osteoclastogenesis previously shown to be produced by osteogenic cells (EPHD4, FASL, and SEMA3A). Although the assays used could not detect the presence of EFNB2 and FASL in the media, all the remaining factors showed increased levels in cultures with induced Gα_s_^R201C^ expression (Figure 5).

### 3.4. Plasmatic Cytokines in FD Patients Correlate with Their Disease Burden

We analyzed 19 cytokines, of which 16 were detectable, in serum samples from 57 patients with FD, with the disease burden ranging from a skeletal burden score (SBS) of 0.5 to 75 (Appendix A). In addition, the bone turnover markers ALP, osteocalcin, and NTX were measured. As expected, ALP, osteocalcin, OPG, and RANKL significantly correlated with patient’s SBS. In addition to these, and for the first time, we demonstrated the positive correlation of seven other cytokines to SBS (Figure 6A). Interestingly, several cytokines correlated with one another and with ALP and osteocalcin, especially all interleukins analyzed, which presented very strong correlations ranging between r = 0.556, *p* = 2.7 × 10^−6^ (IL-6 vs. IL-7) and r = 0.881, *p* = 3.6 × 10^−21^ (IL-7 vs. IL-4) (Figure 6B).

## 4. Discussion

Although our understanding of lesional cell population dynamics in FD pathogenesis has advanced significantly in recent years, we still have limited knowledge of the transcriptional effects of hyperactive Gα_s_ and cAMP excess in BMSCs, the underlying causes of the lesions. Previous differential gene expression analyses of bulk FD tissue fail to capture BMSC-specific transcriptomic changes [11,16,17], and attempts to characterize the effects of Gα_s_^R201C^ expression in human BMSCs through lentiviral transduction involved high cell manipulation, limiting the models’ validity to emulate the transcriptomic profile of FD BMSCs [19,20]. In the present study, we performed a comprehensive exploration of the transcriptome and secretome of FD BMSCs cultured in the absence of other cell types to determine the cell-intrinsic effects of Gα_s_ activation. For a robust experimental design, we combined human and mouse FD cultures to propose a transcriptomic signature comprising significantly modulated orthologous genes changing in the same direction in both models.

For both models, *GNAS* variant expression and cAMP release in the culture media were confirmed before data analysis. Principal component and unsupervised clustering analyses of human and mouse cultures showed good segregation of the samples by group (WT/Gα_s_^R201C^ in mice, HV/FD in humans), and Enrichr analysis identified several terms associated with GPCR/Gα_s_/AC pathway activation, supporting the experimental validity of our approach. In addition, Enrichr analyses of the human or mouse gene lists, as well as the combined FD BMSC signature, undoubtedly supported the importance of cytokine signaling in FD pathogenesis, returning terms like “cytokine activity”, “cytokine-cytokine receptor interaction”, or “inflammatory response” among the top matching genetic datasets. Further analysis of the FD BMSC signature with the STRING protein–protein network analysis tool also confirmed cAMP signaling activation and identified additional expected processes in FD pathogenesis like “extracellular matrix organization”, “metallopeptidase activity”, and “mesenchymal cell proliferation” among others. Lastly, we curated a subset of 57 genes among the 764 identified in the FD Signature with biological significance in the disease (Figure 2B) and demonstrated relationships with cAMP signaling, SSC differentiation fate, osteoclast recruitment, fibrous matrix deposition/remodeling, and vascularization. This restrictive strategy to generate a signature from human and murine cultures likely missed some genes involved in FD BMSC pathogenesis due to the absence of known gene orthologs between species (10–12% of the genes modulated in each dataset lacked an ortholog in the other species). However, at the expense of losing sensitivity, our analysis gained specificity because the 764 genes identified were regulated by *GNAS*’s gain of function in BMSCs across models, avoiding model-specific biases or artifacts. Although human lesional BMSC cultures best represent lesional BMSC transcriptomes, the need to use healthy volunteer BMSC control cultures to compare gene expression introduces several biases. First, BMSCs were obtained from different skeletal sites and through different techniques (iliac crest aspirates in HV versus femur surgical waste tissue in FD patients). In addition, four out of six patients with FD that donated surgical tissue were children, who cannot be age-matched to HVs. Last, Gα_s_^R201C/H^ mRNA allele frequency in FD BMSC cultures was highly variable (7–55%). This biological variability was reflected in our failure to show significant differences in secreted factors between FD and HV BMSCs. On the other hand, mouse-derived BMSC cultures impeccably account for these sources of variability, as paired comparisons can be made between cultures from the same donor, with one induced to express Gα_s_^R201C^. However, they are a less faithful representation of FD BMSCs as transgene expression is not driven by an endogenous promoter and expression is only induced for 48 h, making it impossible to account for the consequences of long-term Gα_s_ activation. To build upon our findings, future studies involving techniques like spatial transcriptomics and single cell RNA sequencing may constitute useful approaches to study FD BSMCs.

Our transcriptomic analysis highlighted the important influence of *GNAS* variant-bearing BMSCs in the local microenvironments of FD lesions, ultimately leading to their characteristic hallmarks. Indeed, as patients age, the number of BMSCs bearing *GNAS* variants decreases, and the cellular composition of FD lesions normalizes [28]. We and others previously pointed to the importance of RANKL in fibrous dysplasia [7], which lead to the development of targeted therapies, first in preclinical and compassionate treatment case studies [8,9,29,30,31] and then in a clinical trial [10,11], including ongoing studies (https://clinicaltrials.gov/study/NCT05419050 [accessed 26 February 2024]). While early studies on the role of RANKL signaling in FD facilitated this promising therapeutic approach, little exploration has been conducted to find additional cytokines and factors secreted by FD BMSCs that may affect osteoclastic–osteoprogenitor crosstalk and other important features of the FD pathogenesis such as fibrous tissue deposition and remodeling, angiogenesis, and nociception. Therefore, we carried out a systematic exploration of cytokines and other selected secreted factors in our culture models.

Due to the biological variability of human FD and HV BMSC cultures, we failed to identify significant changes in their media concentrations of the 18 cytokines analyzed despite the several cytokine-related signatures identified through the Enrichr analyses of the RNAseq data. However, measurements in murine culture media captured a pro-osteoclastic, pro-inflammatory secretome, showing increased levels in 13 of the 25 cytokines examined. RANKL and IL-6 had already been associated with FD pathogenesis [7,32] although a clinical trial using tocilizumab to inhibit IL-6 signaling failed to produce significant disease improvement [15]. Previous analyses failed to detect IL-1α and IL-1β, key pro-inflammatory cytokines, in human FD BMSC cultures [13], but both were detectable in the FD mouse cultures, with IL-1β concentration significantly increased in Gα_s_^R201C^-expressing BMSCs. In addition to these cytokines previously studied in FD, increased levels of IL-2, IL-3, IL-5, IL-12 (p40), IL-13, GM-CSF, IFN-γ, CXCL-1, MCP-1, and TNF-α were found for the first time in the media from Gα_s_^R201C^-expressing BMSCs. We then analyzed both the concentrations of 18 cytokines as well as bone turnover markers known to correlate with disease burden in plasma from patients with FD. Six cytokines showed correlations with disease burden (IL-2, IL-4, IL-7, IL-8, TNF-α), and additional correlations of cytokines to bone turnover markers and to one another were identified, confirming the role of pro-inflammatory cytokines in FD pathogenesis.

Analyzing the individual contribution of these novel cytokines to FD pathogenesis is challenging. In many cases, their role in osteoclastogenesis is equivocal, depending on length of exposure, concentration, and/or combination with other local factors. For example, both pro- and anti-osteoclastogenic properties have been identified for anti-inflammatory cytokines IL-4 and GM-CSF [33,34]. Nevertheless, some of the cytokines identified in our study have unequivocal pro-osteoclastogenic activity. IL-1, IL-7, IL-8, and TNF-α can directly stimulate osteoclastogenesis independently of RANKL [35,36,37,38]. In addition to osteoclastogenesis, cytokines influence other key aspects of the FD lesion microenvironment like angiogenesis [39] and nociceptive pain through local innervation [39] and nociceptive pain through local innervation [40]. Although there are inhibitory drugs targeting several of these cytokines, it is unclear whether this strategy would be useful in FD. First, although cytokine signaling is a local feature of FD, patients do not demonstrate systemic inflammation. Second, inhibiting signaling pathways so broadly involved in human physiology may entail unwanted secondary effects. Anti-RANKL therapy impairs osteoclastogenesis with high specificity and is demonstrated to improve FD lesions without significant negative systemic effects other than impaired bone resorption. However, its discontinuation results in the rebounding of osteoclastic activity. Targeting upstream cytokines like TNF-α may lead to a more gradual, partial inhibition of osteoclastogenesis, potentially preventing a discontinuation rebound, and would likely have direct anti-fibrotic effects on FD lesions. However, TNF-α is involved in a myriad of physiological processes, and systemically inhibiting it comes with several unwanted effects like immunosuppression, increased cancer risk, and others [41,42].

Six additional secreted factors were identified in the media of mouse FD BMSCs. Dkk-1 is a Wnt inhibitor that has been shown to promote osteoclast activity and prevent osteoblast differentiation in response to TNF-α [43]. VEGF, an important pro-angiogenic factor, is also upregulated by TNF-α, and has anabolic effects both in osteoclasts and osteogenic cells [44]. β-NGF, a neurotrophic factor that is also sensitive to pro-inflammatory microenvironments, promotes axonal outgrowth in nearby nociceptive neurons leading to pain [45]. SEMA3A has also been proposed as both a major mediator of nerve regulation in bone turnover as well as a factor in osteoblast–osteoclast coupling [46,47]. Lastly, MMP-2 and FAP-α are proteases involved in remodeling fibrotic tissue whose mRNA levels were significantly upregulated in previous studies and may be useful as biomarkers for FD [11,17,20,48,49,50]. Moreover, these pathways in have been investigated as potential therapeutic targets in a broad range of disorders as fibrosis is a common feature underpinning many disease processes including heart failure [51], cancer [52], diabetes [53], and others [54,55]. The recognition of potential shared pathogenic mechanisms thus raises the possibility of accelerating research in FD/MAS by expanding or repurposing agents developed for treatment in more common disorders. In addition, fibrosis-associated proteases could be leveraged as pro-drug activators targeting pathways like TNF-α that would otherwise give rise to unwanted systemic effects [56].

In conclusion, our study provided a comprehensive analysis of the transcriptomic signature and relevant secreted factors of abnormal BMSCs in FD. By combining data from human and mouse FD models, we identified a robust FD genetic signature, supporting the association of Gα_s_ pathway activation in these cells with cytokine signaling and extracellular matrix reorganization. Our FD secretome profiling identified several pro-inflammatory, pro-osteoclastic cytokines that can be detected in circulation in association with disease severity and other factors relevant in FD pathogenic processes like vascularization, fibrosis, osteoblast–osteoclast crosstalk, and nociception. These findings open novel diagnostic and therapeutic avenues that may be explored in future studies.

## Figures and Tables

**Figure 1 cells-13-00774-f001:**
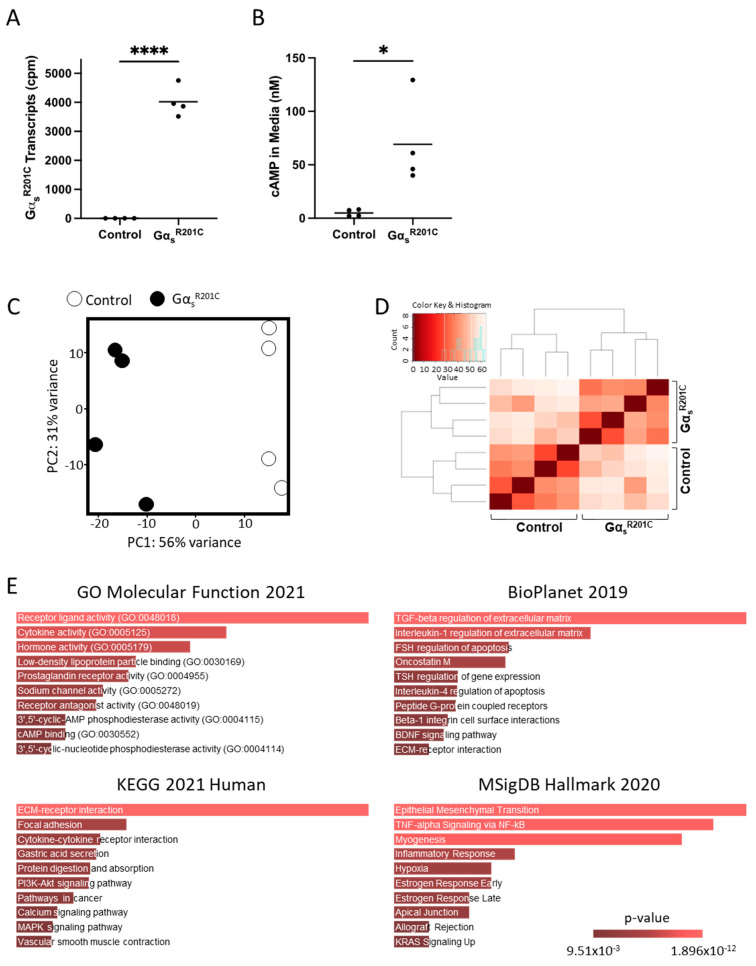
Pro-inflammatory pathways are enriched in BMSCs derived from an inducible mouse model of FD. (**A**) Transcriptomic data confirm that Gα_s_^R201C^ is highly expressed by murine BMSCs following induction with doxycycline. (**B**) BMSCs expressing Gα_s_^R201C^ exhibited markedly elevated cAMP concentrations in culture media, as determined by HPLC-MS. (**C**,**D**) Principal component and unsupervised clustering analyses demonstrated close grouping within control and induced experimental groups while maintaining segregation between groups, confirming robust differences in expression profiles. (**E**) Gene set enrichment analyses of significantly regulated genes (|log_2_FC| > 1.5 and *p*-adj < 0.01) revealed enriched pathways for inflammatory signaling, e.g., “cytokine activity” (GO Molecular Function 2021); “TGF-beta regulation of extracellular matrix” and “Interleukin-1 regulation of extracellular matrix” (BioPlanet 2019); “cytokine-cytokine receptor interaction” (KEGG 2021 Human); and “TNF-alpha Signaling via NF-kB” (MSigDB Hallmark 2020). * *p* < 0.05; **** *p* < 0.0001.

**Figure 2 cells-13-00774-f002:**
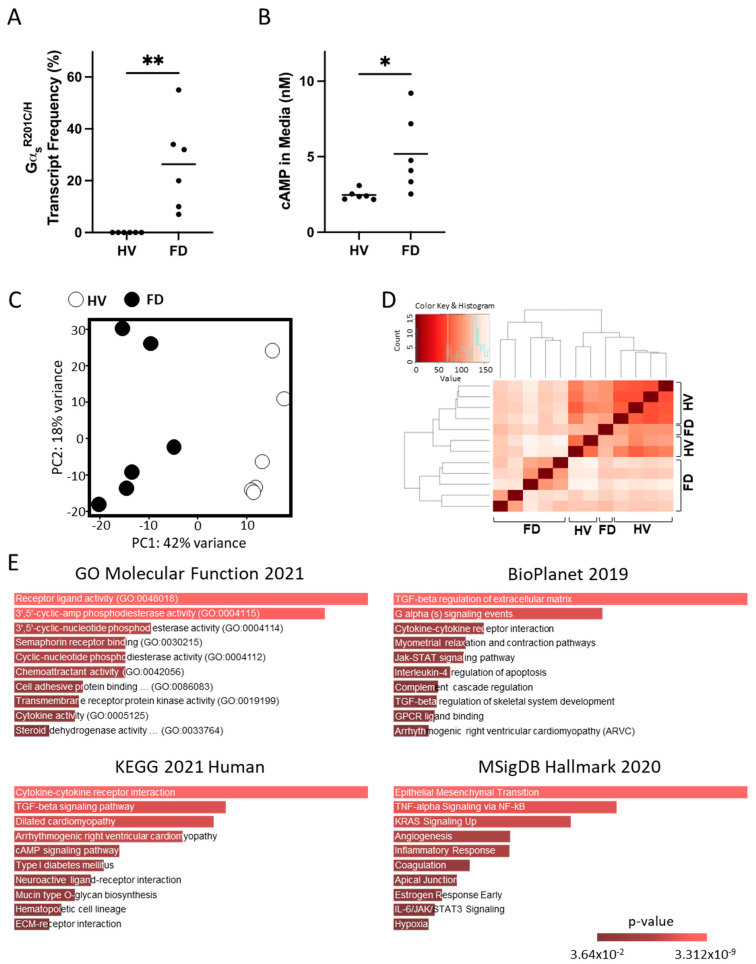
Primary BMSCs from patients with FD also exhibit a pro-inflammatory phenotype. (**A**) Transcriptomic data confirmed expression of Gα_s_ variants in BMSCs from patients with FD. Notably, the fractional abundance of variant transcripts was highly variable, reflecting the mosaic nature of the disease. (**B**) HPLC-MS demonstrated elevated cAMP production in FD cultures compared to healthy volunteers (HVs), in line with Gα_s_ activation. (**C**,**D**) Principal component and unsupervised clustering analyses showed close grouping of HV and FD samples while maintaining segregation between groups. (**E**) Gene set enrichment analyses of significantly regulated genes (|log_2_FC| > 1.5 and *p*-adj < 0.01) revealed enriched pathways for inflammatory signaling, e.g., “cytokine activity” and “chemoattractant activity” (GO Molecular Function 2021); “TGF-beta regulation of extracellular matrix”, “Interleukin-4 regulation of apoptosis”, and “TGF-beta regulation of skeletal system development” (BioPlanet 2019); “cytokine-cytokine receptor activity” (KEGG 2021 Human); and “TNF-alpha Signaling via NF-kB” (MSigDB Hallmark 2020). * *p* < 0.05; ** *p* < 0.01.

**Figure 3 cells-13-00774-f003:**
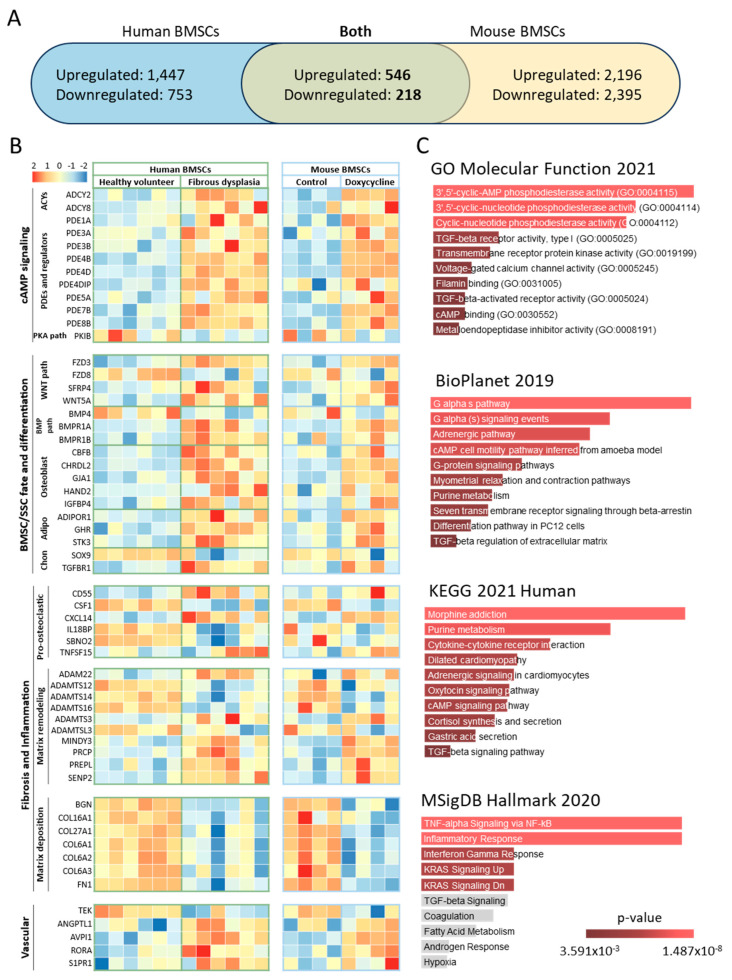
BMSCs from mice and humans with FD shared genes dysregulated in the same manner. (**A**) Differentially expressed genes (*p*-adj < 0.05) were limited to those having orthologs in either species, resulting in 2964 and 5355 differentially expressed human and mouse genes, respectively. Of these, 764 genes appeared in both species and were differentially expressed in the same direction. (**B**) From our FD Signature, genes were manually selected and tabulated according to known associations, e.g., with the WNT and BMP pathways. (**C**) Gene set enrichment analysis of our FD Signature revealed enriched pathways similar to previous results from mice and humans, e.g., “cytokine-cytokine receptor interaction” (KEGG 2021 Human) and “TNF-alpha Signaling via NF-kB” (MSigDB Hallmark 2020). Pathways overlying gray bars did not reach statistical significance. Abbreviations: ACYs, adenylyl cyclases; adipo, adipogenesis; chon, chondrogenesis; PDEs, phosphodiesterases; SSC, skeletal stem cell.

**Figure 4 cells-13-00774-f004:**
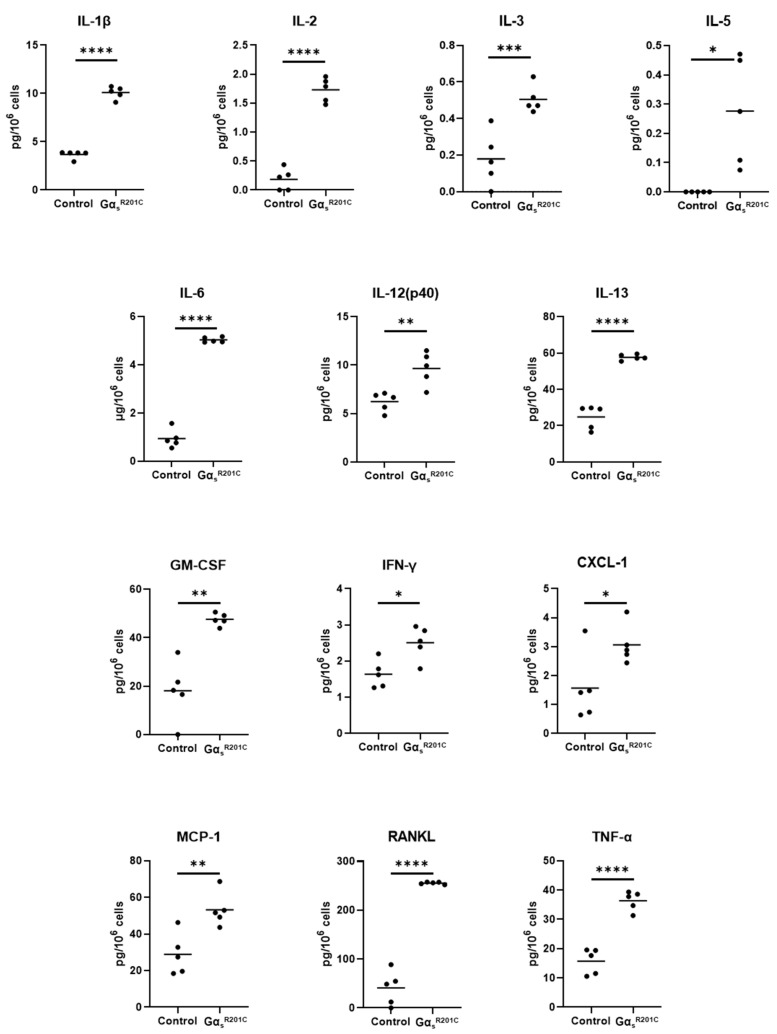
Cultured BMSCs from FD mice release pro-inflammatory cytokines. Multiple Mann–Whitney test with two-stage Benjamini, Krieger, and Yekutieli and 5% FDR was used to determine statistical significance. IL-4 and IL-9 were undetectable. * *p* < 0.05; ** *p* < 0.01; *** *p* < 0.001; **** *p* < 0.0001.

**Figure 5 cells-13-00774-f005:**
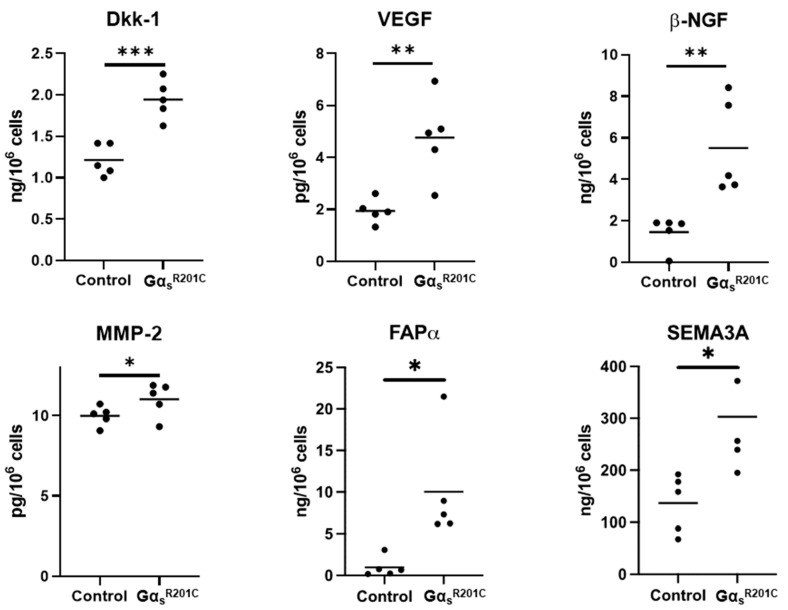
Cultured BMSCs from FD mice release additional factors related to FD pathogenesis. Multiple Mann–Whitney test with two-stage Benjamini, Krieger, and Yekutieli and 5% FDR was used to determine statistical significance. Ephrin B2 and soluble Fas ligand were assessed but not found in detectable levels. * *p* < 0.05; ** *p* < 0.01; *** *p* < 0.001.

**Figure 6 cells-13-00774-f006:**
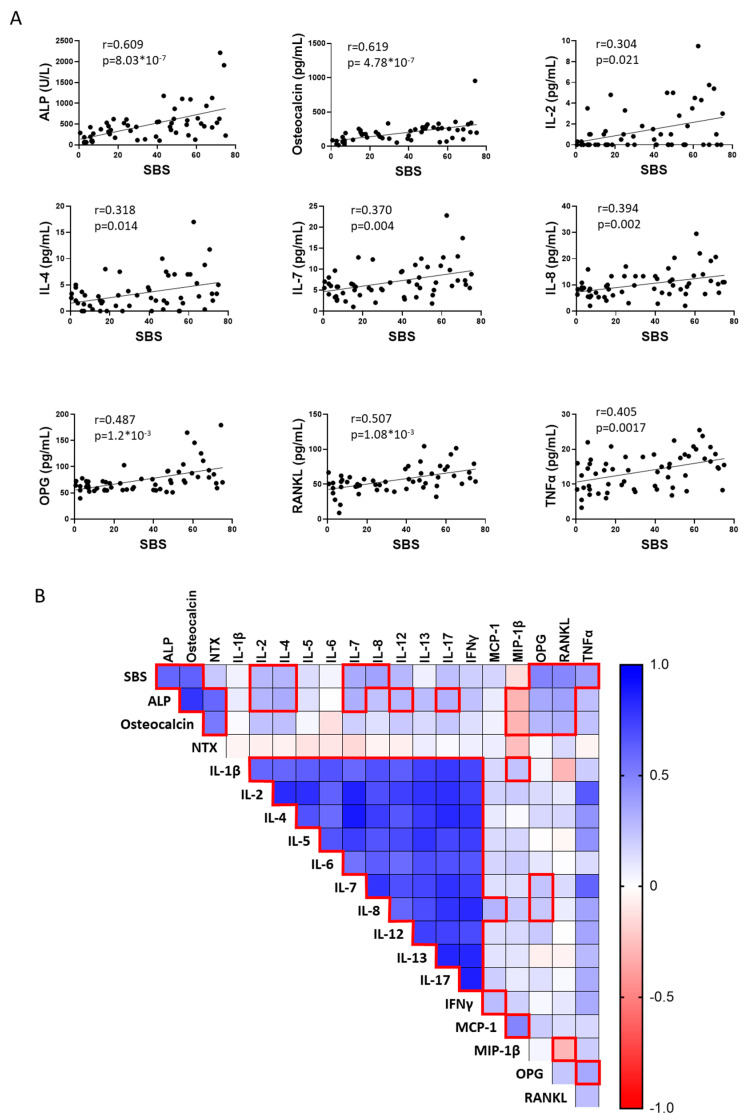
Inflammatory cytokines and bone turnover markers can be detected in serum from patients with FD and correlate with disease burden. (**A**) Cytokines and other markers were analyzed in serum from 57 patients with FD and correlated with skeletal burden score (SBS), a standardized quantification of disease burden. Significance was determined via multiple correlations with Pearson’s r and defined as *p* < 0.05. (**B**) SBS–cytokine and cytokine-cytokine correlations between all detectable factors revealed numerous associations, especially with ALP and osteocalcin, as well as between interleukins as expected (red boxes). *p*-values were adjusted for multiple testing, with an adjusted *p*-value less than 0.05 considered statistically significant. IL-10, G-CSF, and GM-CSF were undetectable.

## Data Availability

mRNA sequencing datasets are available at the NCBI GEO repository under series accession number GSE261360.

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
