# Peer review of "Transcriptomic Signature and Pro-Osteoclastic Secreted Factors of Abnormal Bone-Marrow Stromal Cells in Fibrous Dysplasia"

_cells, 2024, doi:10.3390/cells13090774_

Round 1
Reviewer 1 Report
Comments and Suggestions for Authors
The manuscript provides a comprehensive overview of the research conducted on Fibrous Dysplasia (FD) and its pathophysiological aspects, utilizing both murine and human models to elucidate the role of pro-inflammatory transcriptomic profiles in the disease's progression. Here are some minor suggestions/comments:
1. The introduction provides a solid foundation for understanding Fibrous Dysplasia (FD) and its genetic underpinnings. However, it could benefit from a brief discussion on the prevalence of FD and its impact on patients' quality of life. This addition would contextualize the importance of the study further.
2. The methods section is well-detailed but could improve by providing more information on the criteria for selecting FD patients and healthy volunteers, including age, gender, and other demographic information, to understand the sample population better.
3. The manuscript could benefit from a more detailed discussion on the implications of the identified pro-inflammatory cytokines and factors related to FD pathophysiology. Specifically, exploring their potential role in bone remodeling and fibrous tissue deposition would be insightful.
4. The discussion concludes with the identification of several cytokines and secreted factors as potential therapeutic targets. It would be beneficial to provide specific examples of how some of these targets could be modulated and propose a framework for how these findings might translate into clinical trials or therapeutic interventions.
Comments on the Quality of English Language1. There are minor grammatical errors and awkward phrasings throughout the manuscript that could be revised for clarity. For instance, ensuring consistent tense and clearer sentence structures would improve readability.
2. Ensure that all abbreviations are explained when first used. For instance, "GNAS variant expression and cAMP release in the culture media were confirmed..." could be clearer if "cAMP" is defined as "cyclic Adenosine Monophosphate (cAMP)" upon first mention.
3. Some sentences are quite dense with information and could be broken down for clarity.
Author Response
Dear reviewer 1,
Thank you for taking your time to do a thorough review of our manuscript, which we believe has improved significanlty thanks to your comments and suggestions. Please find below in line explanations on how they were addressed. Your text was bolded to facilitate read. Our edits in the manuscript are highlighted in yellow.
The manuscript provides a comprehensive overview of the research conducted on Fibrous Dysplasia (FD) and its pathophysiological aspects, utilizing both murine and human models to elucidate the role of pro-inflammatory transcriptomic profiles in the disease's progression. Here are some minor suggestions/comments:
Please find below explanations on how your comments were addressed. We appreciate the thorough revision of the manuscript.
- The introduction provides a solid foundation for understanding Fibrous Dysplasia (FD) and its genetic underpinnings. However, it could benefit from a brief discussion on the prevalence of FD and its impact on patients' quality of life. This addition would contextualize the importance of the study further.
Thank you for bringing this up. The introduction has now been modified to include these points (from line 43)
- The methods section is well-detailed but could improve by providing more information on the criteria for selecting FD patients and healthy volunteers, including age, gender, and other demographic information, to understand the sample population better.
The inclusion criteria and demographic information of both the BMSCs and plasma sample donors were included in the material and methods section (Line 109) and supplementary tables S1 and S2. It seems that the supplementary tables were not included in the files sent for revision, we could not find them nor the excel file with tables S3 and S4. We now included S1 and S2 in the main text of the reviewed manuscript.
- The manuscript could benefit from a more detailed discussion on the implications of the identified pro-inflammatory cytokines and factors related to FD pathophysiology. Specifically, exploring their potential role in bone remodeling and fibrous tissue deposition would be insightful.
Additional comments were added to the discussion about the potential effects of the cytokines identified, alongside new reference articles, were included in the discussion (lines 470-478)
- The discussion concludes with the identification of several cytokines and secreted factors as potential therapeutic targets. It would be beneficial to provide specific examples of how some of these targets could be modulated and propose a framework for how these findings might translate into clinical trials or therapeutic interventions.
We now addressed this question between lines 478 and 491.
Comments on the Quality of English Language
- There are minor grammatical errors and awkward phrasings throughout the manuscript that could be revised for clarity. For instance, ensuring consistent tense and clearer sentence structures would improve readability.
- Ensure that all abbreviations are explained when first used. For instance, "GNAS variant expression and cAMP release in the culture media were confirmed..." could be clearer if "cAMP" is defined as "cyclic Adenosine Monophosphate (cAMP)" upon first mention.
- Some sentences are quite dense with information and could be broken down for clarity.
The manuscript language was extensively edited to address these points. Edits are highlighted through the manuscript. Thank you.
Reviewer 2 Report
Comments and Suggestions for Authors
The author has done exemplary work, effectively focusing on the analysis and differentiation of the transcriptome and secretome of bone marrow mesenchymal stromal cells from patients with fibrous dysplasia, employing RNA sequencing and immunoassay techniques. Despite challenges with cytokine analysis due to biological variations, the use of mouse models adeptly filled in the research gaps. The insightful discussion section skillfully navigates these complexities, offering a deep dive into potential cytokine targets for therapeutic intervention. The study offers insightful information on potential target cytokines that could play a crucial role in fibrous dysplasia and potentially serve as therapeutic targets. The clarity and depth of the research significantly contribute to the field, offering valuable insights and directions for future studies.
Suggestions for Improvement:
Fig 2D: The color code box's text is too small, becoming pixelated when zoomed in. It needs better visibility for clarity.
Fig 2E and 3C: The meaning of the graph's colors requires clarification. Without a scale or indication, it's uncertain whether all the pathways shown are significant or only those mentioned in the figure legend. Providing clarification on this detail would significantly enhance the understanding and interpretation of the data presented.
Author Response
Dear reviewer 2,
Thank you for taking your time to do a thorough review of our manuscript, which we believe has improved significanlty thanks to your comments and suggestions. Please find below in line explanations on how they were addressed. Your text was bolded to facilitate read.
The author has done exemplary work, effectively focusing on the analysis and differentiation of the transcriptome and secretome of bone marrow mesenchymal stromal cells from patients with fibrous dysplasia, employing RNA sequencing and immunoassay techniques. Despite challenges with cytokine analysis due to biological variations, the use of mouse models adeptly filled in the research gaps. The insightful discussion section skillfully navigates these complexities, offering a deep dive into potential cytokine targets for therapeutic intervention. The study offers insightful information on potential target cytokines that could play a crucial role in fibrous dysplasia and potentially serve as therapeutic targets. The clarity and depth of the research significantly contribute to the field, offering valuable insights and directions for future studies.
We appreciate these positive comments. Your specific suggestions are addressed below.
Suggestions for Improvement:
Fig 2D: The color code box's text is too small, becoming pixelated when zoomed in. It needs better visibility for clarity.
We appreciate this comment and modified the figure accordingly.
Fig 2E and 3C: The meaning of the graph's colors requires clarification. Without a scale or indication, it's uncertain whether all the pathways shown are significant or only those mentioned in the figure legend. Providing clarification on this detail would significantly enhance the understanding and interpretation of the data presented.
Thank you for bringing this up. A color scale is now included as well as additional text in the figure legend to clarify.